A new Lower Triassic ichthyopterygian assemblage from Fossil Hill, Nevada

Kelley Neil P. 1 2 kelleynp@si.edu
Motani Ryosuke 2
Embree Patrick 3
Orchard Michael J. 4
1 Department of Paleobiology, National Museum of Natural History, Smithsonian , Washington, District of Columbia , United States
2 Department of Earth and Planetary Sciences, University of California, Davis , Davis, California , United States
3 Orangevale, California , United States
4 Natural Resources Canada–Geological Survey of Canada , Vancouver, British Columbia , Canada
Farke Andrew
Electronic publication date: 2016 Jan 26
Publication date: 2016
Volume: 4
Electronic Location ID: e1626
Received 2015 Oct 23; Accepted 2016 Jan 5
Copyright: © 2016 Kelley et al.
Copyright year: 2016
Copyright holder: Kelley et al.
License: This is an open access article distributed under the terms of the Creative Commons Attribution License, which permits unrestricted use, distribution, reproduction and adaptation in any medium and for any purpose provided that it is properly attributed. For attribution, the original author(s), title, publication source (PeerJ) and either DOI or URL of the article must be cited.
License URL: https://creativecommons.org/licenses/by/4.0/

Keywords: Triassic, Ichthyosaur, Ichthyopterygia, Marine reptile, Nevada

Funding: The authors received no funding for this work.

==============================
We report a new ichthyopterygian assemblage from Lower Triassic horizons of the Prida Formation at Fossil Hill in central Nevada. Although fragmentary, the specimens collected so far document a diverse fauna. One partial jaw exhibits isodont dentition with blunt tipped, mesiodistally compressed crowns and striated enamel. These features are shared with the Early Triassic genus Utatsusaurus known from coeval deposits in Japan and British Columbia. An additional specimen exhibits a different dentition characterized by relatively small, rounded posterior teeth resembling other Early Triassic ichthyopterygians, particularly Grippia. This Nevada assemblage marks a southward latitudinal extension for Early Triassic ichthyopterygians along the eastern margin of Panthalassa and indicates repeated trans-hemispheric dispersal events in Early Triassic ichthyopterygians.

Introduction

Ichthyosaurs were among the most enduring and successful Mesozoic marine reptile groups, appearing in the Early Triassic and persisting some 150 million years until their extinction in the Late Cretaceous (McGowan & Motani, 2003). The fossil record of early ichthyopterygians (the clade comprising ichthyosaurs and close relatives) includes a variety of morphologically disparate taxa from widespread localities in Asia, North America and the Arctic. Most of these assemblages are broadly contemporaneous, all being late Spathian (late Early Triassic) in age. Recent discoveries in China (Ji et al., 2014; Motani et al., 2015) have extended this record earlier into the Spathian and have shed new light on the phylogenetic and biogeographic origins of the clade. However, the rapid early diversification and trans-hemispheric dispersal history of ichthyopterygians during the Early Triassic remains poorly understood.

Nevada has been an important source of Triassic marine reptile fossils since the 19th Century producing abundant and well-preserved Middle Triassic (Leidy, 1868; Merriam, 1905; Merriam, 1908; Merriam, 1910; Sander, Rieppel & Bucher, 1994; Sander, Rieppel & Bucher, 1997; Fröbisch, Sander & Rieppel, 2006; Fröbisch et al., 2013) and Late Triassic (Camp, 1976; Camp, 1980) ichthyopterygian and sauropterygian fossils. Notably, Early 20th Century field work led by John Merriam and Annie Alexander at the Fossil Hill locality in the Humboldt Range produced several specimens of the ichthyosaur Cymbospondylus (Merriam, 1908)–previously described by Leidy (1868) on the basis of fragmentary remains–as well as the type specimens of Omphalosaurus nevadanus (Merriam, 1906) and Phalarodon fraasi (Merriam, 1910). Later work by Camp (1976); Camp (1980); Sander, Rieppel & Bucher (1994); Sander, Rieppel & Bucher (1997) and Schmitz et al. (2004) and others illuminated rich Middle and Late Triassic marine reptile assemblages preserved in Nevada.

In contrast, knowledge of Early Triassic marine reptile fossils in this region is scant. The only published Early Triassic marine reptile occurrence from Nevada is based on a partial jaw referred to the enigmatic genus Omphalosaurus and described as a second species, O. nettarhynchus (Mazin & Bucher, 1987). This specimen was collected from the Spathian-aged informally designated “lower member” of the Prida Formation in the Humboldt Range, which sits immediately below the well-known Fossil Hill Member of the Prida Formation, famous for its rich marine reptile assemblage including the aforementioned Cymbospondylus, Phalarodon, and Omphalosaurus nevadanus. Fragmentary, float-derived remains of Early Triassic ichthyopterygians have been reported from Spathian horizons in southeastern Idaho (Massare & Callaway, 1994; Scheyer et al., 2014) roughly 500 km to the northeast of the Fossil Hill locality. Even further to the east, the early sauropterygian Corosaurus alcovensis (Case, 1936) is known from the Alcova limestone in Wyoming whose Early Triassic age was recently confirmed (Lovelace & Doebbert, 2015).

Here, we report a new Early Triassic ichthyopterygian assemblage from the lower member of the Prida Formation at the Fossil Hill Locality. These fossils are Spathian (Lower Triassic) in age based on co-occuring conodont and ammonoid faunas and sit stratigraphically below the diverse Middle Triassic marine reptile assemblage from the Fossil Hill Member of the Prida Formation. These occurrences extend the southward latitudinal range of early ichthyopterygians in North America and demonstrate that early in their evolutionary history, multiple ichthyopterygian taxa quickly dispersed around or across wide expanses of ocean and ranged from sub-tropical to high temperate waters on the eastern margin of northern Panthalassa.

Institutional Abbreviations

USNM, National Museum of Natural History, Smithsonian Institution, Washington, D.C., U.S.A.

Materials and methods

Geological and stratigraphic setting

The new fossils reported here were collected from multiple horizons within the unnamed lower member of the Prida Formation of the Star Peak Group at Fossil Hill, on the eastern flank of the Humboldt Range in Pershing County, Nevada (Fig. 1). The Star Peak Group consists of a sequence of syndepositionally deformed carbonate-dominated units deposited on what was then the western shelf of North America (Nichols & Silberling, 1977; Wyld, 2000). In the study area, the lower member of the Prida Formation forms the base of the Star Peak Group and sits unconformably atop the Permian/Lower Triassic aged Koipato Group volcanics (Wyld, 2000).

Figure 1 Summarized stratigraphy and regional map.

(A) Stratigraphy of the Triassic Prida Formation near Fossil Hill in the Humboldt Range, Nevada indicating horizons of specimens USNM 559349 and 559350 and conodont samples. (B) Regional map, modified from Silberling (1962).

The lower member of the Prida Formation transitions from siliciclastic sand and conglomerate layers near the contact with the underlying Koipato Group to dark-grey limestone above with intermittent microbialite, conglomerate and chert-dominated beds. The presence of conglomerates and microbialites indicate relatively shallow conditions with a general trend towards deeper water facies characteristic of the overlying Fossil Hill Member (Wyld, 2000). Gastropods and bivalves are abundant in lower layers whereas conodonts and ammonoids are found locally within middle and upper layers of the lower member. Scattered vertebrate fossils occur in multiple horizons within the lower member (Fig. 1), but are most abundant in the middle carbonate layers where they are associated with the conodont Triassospathodus symmetricus (Orchard, 1995) and the ammonoid Prohungarites gutstadti (Guex et al., 2010) indicating a late Spathian age (Subcolumbites ammonoid biozone). These fossils were collected by the landowner and co-author Patrick Embree, who donated the material to the Smithsonian National Museum of Natural History.

Results

Systematic paleontology

Ichthyopterygia (Owen, 1840).

cf. Utatsusaurus (Shikama, Kamei & Murata, 1978).

Diagnosis. Teeth smaller than marginal dentition present on pterygoid; squamosal not entirely eliminated from supratemporal fenestra by supratemporal; interclavicle cruciform; dorsal margin of external naris formed entirely by nasal; prefrontal shelf prominent; transverse flange of the pterygoid well defined and anterolaterally projecting; supratemporal terrace present; tooth implantation subthecodont, with both dental groove and shallow socket; tooth crowns of middle to posteriorly placed teeth distomesially compressed; humerus as wide proximally as distally; humerus longer than wide; ulnar facet of humerus as wide as radial facet; no more than five phalanges in any digit; posterior dorsal vertebrae cylindrical in outline (from Cuthbertson, Russell & Anderson, 2013a; after Motani, 1999; McGowan & Motani, 2003).

Referred specimen. USNM 559349 Partial mandible including teeth. (Fig. 2).

Figure 2 Specimen USNM 559349, partial ichthyopterygian jaw cf. Utatsusaurus.

(A) Complete specimen, in labial view, anterior to the left. Squares on scale bar equal 5 mm. (B) Magnified view of anterior dentition, squares on scale bar equal 1 mm.

Locality. Fossil Hill, Humboldt Range, Pershing County, Nevada.

Horizon and age. Found as surface float within an outcrop of Lower Triassic (upper Spathian) lower member of Prida Formation, Star Peak Group. Based on location and matrix lithology this jaw is inferred to derive from horizon FH1–7 (Fig. 1), which is Spathian based on the occurrence of the ammonoid Prohungarites gutstadi (i.e. Subcolumbites Zone of Guex et al. (2010)) and conodonts Triassospathodus symmetricus (Orchard, 1995) and Neostrachanognathus sp. extracted from the matrix.

Description. USNM 559349 is a partial mandible measuring 82 mm long. The jaw fragment preserves portions of the dentary, surangular and splenial. The surfaces of the dentary and surangular are heavily striated and the orientation of these striations differs between the bones. The suture between the dentary and surangular is long and straight, extending across the entire preserved portion of the jaw. In places this suture is indistinct but can be traced by the contrasting surface striation patterns of the dentary and surangular. The splenial can be observed at the broken anterior edge of the fossil where it comprises the medial and ventral portion of the jaw where a thin projection wraps underneath the surangular. A row of irregular weathered depressions follows the approximate course of the suture between the surangular and dentary but it is not clear if these represent natural foramina or are simply artifacts of weathering. Judging from the arrangement of the bones, the fragment likely represents a central-posterior portion of the left mandibular ramus anterior to the coronoid process.

Thirteen lower teeth are present, along with an additional poorly preserved isolated tooth between the tenth and eleventh in-place teeth, which may be either a disarticulated upper or lower tooth. The teeth are set within alveoli along a continuous groove. No distinct bony septa between alveoli are visible but may be present at the bottom of the dental groove, being concealed in matrix that is very difficult to remove through mechanical preparation. The roots of some teeth are clearly expanded at the base and exhibit plications that are coarser than crown striation. The most anterior tooth is completely exposed anteriorly, revealing its root structure inside the dental groove. It is seen there that the root ceases its expansion once inside the groove, and teeth are embedded to both the labial wall and the base of the groove. A narrow gap emerges between the lingual wall and the root toward the dentigerous margin. Tooth implantation is likely subthecodont (sensu Motani, 1997a), although histological study is necessary to firmly establish this. The root cross-section is much wider than long, as reported for Utatsusaurus hataii (Motani, 1996).

Teeth are isodont and conical with striated crowns. Tooth roots are extensively exposed above the alveolar margin such that they account for half or more of the exposed height of each tooth. Tooth crowns are distinguished from these exposed roots by a distinct margin, with most crowns slightly constricted at their base. Some crowns exhibit slightly higher convexity of their anterior surface relative to the posterior surface given them a slightly recurved appearance. The teeth are also recurved lingually toward the tip, as clearly seen in the most anterior tooth (Fig. 2B). This curvature closely resembles what was described for Utatsusaurus hataii (Motani, 1996:Fig. 3; Cuthbertson, Russell & Anderson, 2013a:Figs. 7C and 7D). The tips of the teeth are relatively blunt. Tooth crowns are approximately 3.6 mm tall and 2 mm in mesiodistal diameter. Spacing between teeth ranges from 3 mm to 6 mm, more widely spaced teeth may have replacement teeth between them. One 10 mm gap along the tooth row likely represents at least one missing tooth. Several teeth are broken, either at the root or the crown, revealing a pulp cavity without evidence of infolding of the dentine.

Remarks. The tooth morphology observed in this specimen closely resembles that described for Utatsusaurus hataii from the Lower Triassic Osawa Formation of Kitakami, Japan (Shikama, Kamei & Murata, 1978). Most notably, the teeth curve lingually and slightly posteriorly toward the tip, which is a feature that is uniquely known in Utatsusaurus among basal ichthyopterygians. Other shared features include: isodont dentition, tall exposed roots, blunt conical striated crowns, a slightly constricted base of some crowns, and an absence of infolding in the pulp cavity. Each of these features is observed in the holotype of Utatsusaurus hataii (IGPS 95941) (Motani, 1996) and most are reported in a referred specimen (UHR 30691) (Cuthbertson, Russell & Anderson, 2013a). No other Early Triassic or later ichthyopterygian exhibits this suite of dental characters.

The teeth of this specimen do diverge from Utatsusaurus hataii in their much larger size. The maximum tooth exposed height in this specimen is 11 mm, whereas the maximum crown height and width are 4.5 mm and 2.3 mm respectively (Table 1), compared with 3.3 mm, 1.7 mm and 0.9 mm for the same measurements of teeth in the holotype of Utatsusaurus hataii (Motani, 1996). However, the holotype represents a juvenile (Motani, 1997c), so the size difference may partly be explained as ontogenetic variation. A referred specimen (UHR 30691) is somewhat larger than the holotype, however, the teeth of this specimen are still considerably smaller than those of USNM 559349. Despite this difference in size, the overall shape of the teeth in each of these specimens is very similar. Motani (1996) reported “crown shape index” values–calculated as crown height divided by average basal diameter of the tooth crown (Massare, 1987)–ranging between 0.9 and 3.1 with an average value of 1.9 in the type specimen of Utatsusaurus hataii, while these values range from 1.3 to 2.3 with an average of 1.8 in USNM 559349. Likewise, the “crown ratio”–calculated as crown height divided by total exposed tooth height–averages 0.51 in IGPS 95941 and 0.43 in USNM 559349.

Table 1 Summarized tooth measurements from USNM 559349 and USNM 559350.

All measurements in mm except for shape index and crown ratio, which are ratios.

Specimen		Proximal width (mm)a	Exposed height (mm)b	Crown width (mm)c	Crown height (mm)d	Crown shape indexe	Crown ratiof	
USNM 559349	Max.	4.2	11.1	2.3	4.5	2.3	0.51	
	Min.	2.5	6.4	1.9	2.7	1.3	0.39	
	Mean	3.2	8.5	2.1	3.6	1.8	0.43	
USNM 559350	Max.	1.6	2.5	2.1	1.7	2.1	0.88	
	Min.	1.0	1.8	1.2	1.1	1.0	0.61	
	Mean	1.4	2.2	1.6	1.5	1.4	0.69	
Notes:

a Measured as mesio-distal width of the root at the jawline, following Motani (1996).

b Measured as distance from tip of crown to jaw margin.

c Measured as mesio-distal width of the crown at its widest point.

d Base of crown is distinctive in USNM 559349 due to crown ornamentation; base of crown in USNM 559350 is less distinct but can be approximated by slight basal constriction.

e Calculated as crown height/crown width, following Massare (1987).

f Calculated as crown height/exposed tooth height, following Motani (1996).

The overall arrangement of bones in the fragmentary mandible is consistent with that observed in the Utatsusaurus holotype (IGPS 94941) and the referred specimen (UHR 30691), notably the long, straight contact between the dentary and surangular. The suture between the surangular and dentary is well developed in IGPS 94941 but indistinct in the larger UHR 30691, possibly a consequence of differing ontogenetic stages, or differential weathering, between these two specimens. USNM 559349 approaches UHR 30691 in that the suture between the dentary and surangular is somewhat obscured, but its approximate course can be traced by a contrast in bone texture and an irregular row of weathered pits. This might suggest a relatively mature individual, as has been proposed for IGPS 9494, but this is highly speculative given the incomplete nature of the material.

A partial skull from the Lower Triassic Vega Phroso Member of the Sulphur Mountain Formation from British Columbia, Canada was referred to Utatsusaurus sp. by Nicholls & Brinkman (1993), based largely on the presence of the same dental features detailed above. The teeth of the British Columbia specimen are similar in size to those of the Nevada specimen described here (Nicholls & Brinkman, 1993) and distinctly larger than those found in the holotype of Utatsusaurus hataii from Japan (Motani, 1996). It is therefore possible that these larger-toothed specimens from the eastern margin of Panthalassa (Nevada, British Columbia) represent a form allied with but distinct from Utatsusaurus hataii; however more complete material is needed before this can be confirmed. Recently, Cuthbertson, Russell & Anderson (2014) described another partial skull from the Vega Phroso Member, which they also referred to Utatsusaurus sp., although they concluded that the material originally referred to this genus by Nicholls & Brinkman (1993) was non-diagnostic at the genus level. Unfortunately this recently described material lacks a lower jaw or teeth and cannot be compared to USNM 559349.

Ichthyopterygia (Owen, 1840).

cf. Grippiidae (Wiman, 1929).

Definition. The last common ancestor of Grippia longirostris and Gulosaurus helmi, and all its descendants (Ji et al., 2015).

Diagnosis. Maxilla with multiple tooth rows; posterior tooth crown rounded; supratemporal-postorbital contact present; proximal manual phalanges not closely packed proximo-distally (from Ji et al. (2015)).

Referred specimen. USNM 559350 Partial maxilla including teeth. (Fig. 3).

Figure 3 USNM 559350 Partial ichthyopterygian maxilla cf. Grippidia.

(A) Partial maxilla in lingual view, anterior to the left. Squares on scale bar equal 1 mm. White arrow indicates possible attachment facet for tooth in second lingual tooth row. (B) Magnified view of dentition.

Locality. Fossil Hill, Humboldt Range, Pershing County, Nevada.

Horizon and Age. Collected from FH1-5 (Fig. 1) horizon which is Spathian in age based on the occurrence of the ammonoid Prohungarites gutstadi (Guex et al., 2010) and conodonts including Triassospathodus and Neostrachanognathus. This horizon is also characterized by distinctive spherical structures originally interpreted as microbial (i.e. ‘oncoids’) but more recently suggested to represent sponge “reefs” (Brayard et al., 2011; see further discussion below).

Description. USNM 559350 (Fig. 3) is a partial maxilla measuring 20 mm and bearing five teeth exposed in medial view. The teeth are robust cones exhibiting a trend of posteriorly increasing basal diameter, whereas the crown height remains constant giving the posteriormost preserved tooth a distinctly rounded shape. There is a distinct constriction below the crown separating it from the root below, however the constriction is very slight in the anteriormost tooth. The anteriormost crown height and width are 1.7 mm and 1.2 mm respectively; in the second posteriormost tooth, which is better preserved than the posteriormost tooth, crown height and width are 1.3 mm and 2.1 mm. The tooth enamel appears smooth and polished with little indication of striation; however, this could be attributed to tooth wear. Faint plication is visible on some roots. Although some teeth are abraded, none expose the pulp cavity clearly enough to determine presence or absence of infolded dentine.

In medial view the teeth are attached to the lingual wall of the maxilla, representing pleurodont tooth attachment. An expanded bone of attachment conceals the bases of the two posteriormost teeth, suggesting subleurodont attachment, a modified form of pleurodont attachment (Motani, 1997a), in at least the posterior region of the maxillary tooth row. While only a single row of teeth is observed, a shallow depression on the lingual margin of the tooth row immediately anterior to the second posteriormost tooth could represent the attachment facet of a missing tooth. If this were the case it might possibly represent a second row immediately lingual to the preserved teeth. Wide spacing between the four anteriormost teeth would easily accommodate an additional offset tooth row as observed in the maxillary dentition of Grippia (Motani, 1997b) and Gulosaurus (Cuthbertson, Russell & Anderson, 2013b).

Remarks. Among Early Triassic ichthyopterygians, small, robust teeth, similar to those reported here, are typical of the posterior dentition of Grippia (Motani, 1997b). Rounded teeth are also observed in the Early Triassic genus Chaohusaurus (Motani & You, 1998) and, to a lesser extent, Gulosaurus (Cuthbertson, Russell & Anderson, 2013b). Grippia was previously reported from the Lower Triassic Vega Phroso Member of the Sulphur Mountain Formation in British Columbia (Brinkman, Xijin & Nicholls, 1992). This specimen was later redescribed as a distinct taxon, Gulosaurus helmi (Cuthbertson, Russell & Anderson, 2013b) and found to be sister taxon to Grippia longirostris. Similarly, recent work by Ji et al. (2015), established a new clade Grippioidea including Grippia, Gulosaurus, Utatsusaurus, and Parvinatator, (Nicholls & Brinkman, 1995) although the precise relationships among these taxa varied somewhat depending on taxon and character inclusion. The enigmatic marine reptile Omphalosaurus nettarhynchus (Mazin & Bucher, 1987), previously reported from Spathian lower member of the Prida Formation, also possesses rounded dentition, but is distinct from this specimen by its much larger size and in exhibiting a broad pavement of rounded teeth on the mandible.

Alternatively, this specimen may have some affinity with Chaohusaurus, an Early Triassic ichthyopterygian from China in which some specimens also show distinctly rounded posterior dentition (Motani & You, 1998). However the posterior teeth of Chaohusaurus are generally smaller and more tightly packed than in USNM 559350, averaging approximately ten teeth over 20 mm rather than the five teeth observed over the same distance in this specimen. Chaohusaurus was previously regarded as a grippidian partly on the basis of possessing multiple maxillary tooth rows and rounded posterior dentition (Motani, 1999). However, more recent analyses (Cuthbertson, Russell & Anderson, 2013b; Ji et al., 2015) do not support this placement.

The crown shape index in USNM 559350 ranges from 1.0–2.1 and averages 1.4 (Table 1), similar to the average shape index reported for the maxillary dentition of Grippia, 1.4 (Motani, 1997b) and Gulosaurus 1.64 (Cuthbertson, Russell & Anderson, 2013b). Thus, the rounded dentition observed in USNM 559350 is strongly suggestive of an affinity with some other Early Triassic ichthyopterygians but more precise placement will require more complete skeletal material.

Discussion

Despite the fragmentary nature of the remains described here, their resemblance with the distinctive dentitions of other Early Triassic ichthyopterygians allows tentative interpretations to be made. The presence of Utatsusaurus-like and Grippia or Chaohusaurus-like forms suggests similarity with the Lower Triassic Vega-Phroso assemblage from the Wapiti Lake region of British Columbia, from which Utatsusaurus (Nicholls & Brinkman, 1993; Cuthbertson, Russell & Anderson, 2014) and grippidians (Brinkman, Xijin & Nicholls, 1992; Cuthbertson, Russell & Anderson, 2013b) have also been reported. However, the type locality of Utatsusaurus is in the Osawa Formation of Japan (Shikama, Kamei & Murata, 1978), whereas the type localities of Grippia and Chaohusaurus are in the Vikinghøgda Formation (= “Sticky Keep Formation” of older references) of Spitsbergen (Wiman, 1929; Wiman, 1933; Hounslow et al., 2008) and the Nanlinghu Formation of Anhui Province, China (Young & Dong, 1972), respectively. Thus, Early Triassic ichthyopterygian taxa were widely distributed around the margins of northern Panthalassa (Cuthbertson, Russell & Anderson, 2013b).

This broad distribution early in their evolutionary history, from numerous Late Spathian (Subcolumbites Zone) localities of broadly coeval age (Scheyer et al., 2014), has made it difficult to pinpoint the biogeographic origins of the group. However, recent work in China has extended the biostratigraphic range of ichthyopterygians to the underlying Procolumbites Zone (Motani et al., 2014; Ji et al., 2014). Furthermore, the occurrence of diverse and endemic hupehsuchians, widely regarded as the ichthyopterygian sister-group, and the plesiomorphic ichthyosauromorph Cartorhynchus (Motani et al., 2015) are consistent with an origin of ichthyopterygians near the south China block in equatorial western Panthalassa.

The inferred nearshore lifestyle of most Early Triassic ichthyopterygians has led others to propose that these early marine reptiles dispersed along coastlines or across transient epicontinental corridors (Cuthbertson, Russell & Anderson, 2013b). However, there is little geological evidence for such corridors in the Early Triassic, which was a time of relatively low global sea level (Miller et al., 2005). Furthermore the absence of Early Triassic ichthyopterygian fossils in Western Tethys is surprising under this scenario. Conversely, the biogeographic histories of other aquatic–and even terrestrial–reptile groups are marked by occasional transoceanic dispersal events (Rocha et al., 2006; Vélez-Juarbe, Brochu & Santos, 2007), and such events could explain the distribution of Early Triassic ichthyopteryians on opposite shores of Panthalassa.

Brayard et al. (2009) identified trans-Panthalassan distribution patterns in Spathian ammonoids, identifying similar ammonoid faunas in Nevada, Kitakami and British Columbia, which they attributed to oceanographic currents. The occurrence of some Early Triassic marine reptile taxa (e.g. Utatsusaurus) on both the eastern and western margins of of Panthalassa might reflect sporadic crossing of deep ocean basins by these lineages, potentially facilitated by the same ocean currents that mediated transoceanic dispersals of contemporaneous marine invertebrates (Fig. 4). The wide distribution of Early Triassic sauropterygians, including the South China Block (Jiang et al., 2014) and western margin of North America (Storrs, 1991; Lovelace & Doebbert, 2015) on opposite shores of Panthalassa indicates to a similar dispersal history in the early members of that marine reptile clade. Isolated terranes such as South Kitakami, South Primoyre and Chulitna could have served as stepping-stones for shallow marine taxa. Dispersal along coastlines around the northern margins of Panthalassa remains an alternative scenario that could explain the broad distribution of some Early Triassic ichthyopterygians, with pronounced global warmth in the Early Triassic mediating limiting climatic conditions at high latitudes (Sun et al., 2012). However, the apparent absence of ichthyopterygian fossils from high latitudes on the western margin of northern Panthalassa remains a puzzle under this scenario.

Figure 4 Distribution of Early Triassic ichthyopterygians.

Paleogeographic distribution of Early Triassic ichthyopterygians, map modified from Brayard et al. (2009). Locality abbreviations as follows: (B) British Columbia; (C) South China; (I) Idaho; (K) South Kitakami; (N) Nevada (present study, highlighted in yellow); (S) Spitsbergen; (T) Timor. Arrows indicate inferred ammonoid dispersal routes (Brayard et al., 2009).

Intriguingly, the oldest marine reptile bearing horizons at Fossil Hill are associated with a prominent limestone marker bed bearing distinctive spherical structures ~1–2 cm in diameter (Fig. 5). We initially interpreted these structures as microbial ‘oncoids.’ Widespread microbialite-dominated facies are characteristic of Lower Triassic strata globally, including in the western United States (Pruss & Bottjer, 2004; Baud, Richoz & Pruss, 2007), and are interpreted as a byproduct of the end-Permian mass extinction and subsequent delayed biotic recovery of metazoan reefs (Pruss & Bottjer, 2004). A similar association between the basal sauropterygian Corosaurus and stromatolites in the Lower Triassic Alcova limestone has been reported previously (Storrs, 1991). More recently, similar spheroidal structures from the Humboldt Range and other localities in western North America have been interpreted as ‘transient sponge reefs’ (Brayard et al., 2011). Thus, the diversification and dispersal of Early Triassic marine reptiles was apparently well underway at the end of the Early Triassic (Scheyer et al., 2014) despite some lingering signs of continued environmental stress preserved in the same strata. Future work at this new locality, and elsewhere, may help to clarify the role that large-scale environmental changes played in shaping the early evolutionary history of Mesozoic marine reptiles.

Figure 5 Distinctive sedimentary structures associated with horizon of USNM 559350.

Spherical structures in FH1-5 that may represent microbial structures or sponges. This appears to be a widespread and distinctive regional Lower Triassic facies associated with recovery from the end-Permian mass-extinction. Vertebrate fossils also occur in this horizon including USNM 559350 described here. Hammer for scale is approximately 30 cm in length.

Supplemental Information

Supplemental Information 1 Summary of conodont results.

Click here for additional data file.

We thank Tetsuya Sato for assistance with preparation of USNM 559349. Torsten Scheyer, Lars Schmitz, and Cheng Ji all provided helpful discussion. Reviewers Robin Cuthbertson and Nadia Fröbisch and the handling editor also provided critiques and suggestions that greatly improved this paper.

Additional Information and Declarations

Competing Interests

Author Contributions

Data Deposition

The authors declare no competing interests.

Neil P. Kelley conceived and designed the experiments, performed the experiments, analyzed the data, contributed reagents/materials/analysis tools, wrote the paper, prepared figures and/or tables, reviewed drafts of the paper.

Ryosuke Motani conceived and designed the experiments, performed the experiments, analyzed the data, contributed reagents/materials/analysis tools, reviewed drafts of the paper.

Patrick Embree conceived and designed the experiments, performed the experiments, analyzed the data, contributed reagents/materials/analysis tools, prepared figures and/or tables, reviewed drafts of the paper.

Michael J. Orchard performed the experiments, analyzed the data, contributed reagents/materials/analysis tools, wrote the paper, reviewed drafts of the paper.

The following information was supplied regarding data availability:

All data associated with this paper is included in the manuscript and the Supplemental Information.

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
