# Peer review of "A new Lower Triassic ichthyopterygian assemblage from Fossil Hill, Nevada"

_PeerJ, doi:10.7717/peerj.1626_

## Round 0.1 · original submission · Minor Revisions

- Both reviewers suggest adding some additional description and comparisons of USNM 559349--this should be given strong consideration during revision.
- Reviewer Cuthbertson recommends adding some additional basic measurements and indices--I agree that both should be added into the revision if possible.
- Both reviewers suggest a variety of more minor grammatical and stylistic edits that should be addressed in revision as applicable.
- What permits or permissions were required and obtained for this study? These must be given here, and the appropriate land agencies or land owners acknowledged as well.

Additional minor edits in addition to those suggested by the reviewers:
line 170: Do you mean "Grippidia" instead of "Grippida"?
line 234: "Grippidians" should be lower case
line 267: add "of" in "the apparent absence ichthyopterygian"

·

Basic reporting

I have no major complaints with the 'Basic Reporting' of the manuscript. However, there are some minor points that should be corrected prior to publication (please see attached pdf), some of which include:

1) Sufficient background information is provided in the Introduction, but more emphasis could/should be provided to highlight the importance of the material (i.e., Early Triassic ichthyopterygian material is rare and this assemblage provides insights into early morphological patterns).

2) Overall, the literature cited is mostly appropriate. However, I would suggest adding the following reference that represents the most recent taxonomic diagnosis/description of Utatsusaurus: Cuthbertson, R., Russell, A., and Anderson, J. 2013. Re-interpretation of the cranial morphology of Utatsusaurus hataii (Ichthyopterygia) (Osawa Formation, Lower Triassic, Miyagi, Japan) and its systematic implications. The Journal of Vertebrate Paleontology 33: 817-830.

3) There are many instances where sentence structure should be revisited - please be sure to use commas where appropriate. In addition to this, I've indicated some grammatical changes in the attached pdf that should improve the overall flow of the ms.

4) For consistency and clarity, I would suggest using either 'ichthyopterygian' or 'ichthyosaur'.

Experimental design

Aspects of the Experimental Design could be improved upon, which would strengthen the utility and research potential of the manuscript.

1) No definitive research question is stated. Providing a research question/hypothesis would provide direction to the manuscript and help highlight the importance of this new Early Triassic assemblage and what knowledge gap it contributes to filling.

2) I would strongly advise the authors to add a table of tooth dimensions for each described specimen, with a calculation of common dental indices (e.g., crown shape index) used in some of the cited papers, including:

Motani, R. 1996. Redescription of the dental features of an Early Triassic ichthyosaur, Utatsusaurus hataii. Journal of Vertebrate Paleontology, 16:396–402.

AND

Cuthbertson, R., Russell, A., and Anderson, J. 2013. Cranial morphology and relationships of a new grippidian (Ichthyopterygia) from the Vega-Phroso Siltstone Member (Lower Triassic) of British Columbia, Canada. The Journal of Vertebrate Paleontology 33: 831-847.

In addition, dental comparisons should be made between USNM 559349 (described here) and UHR30691, representing another specimen of Utatsusaurus (described in the Cuthbertson, R., Russell, A., and Anderson, J. 2013. Re-interpretation of the cranial morphology of Utatsusaurus hataii (Ichthyopterygia) (Osawa Formation, Lower Triassic, Miyagi, Japan) and its systematic implications. The Journal of Vertebrate Paleontology 33: 817-830). Comparison between these specimens is missing from their Remarks section (page Line 141) and should be added.

Validity of the findings

1) Adding the abovementioned dental index will provide an additional means of testing the taxonomic conclusions of the manuscript. For this reason, I recommend they be added to a revised version.

2) My other concern is the conclusions that are made surrounding the early radiation patterns of ichthyopterygians. The abstract states that the assemblage 'indicates repeated circum-oceanic dispersal events in Early Triassic ichthyopterygians', with further elaboration starting on Line 253. Although I can appreciate their perspective on the possibility of trans-oceanic dispersal, no argument is provided to preclude dispersal through epicontinental corridors. As such, based on the current fossil evidence, I find it premature to declare a trans (or circum)-oceanic dispersal pattern for early group members.

Additional comments

To the Authors,

I enjoyed reviewing this manuscript and feel that you have done a good job of highlighting some fragmentary, yet important, material. The paper helps shed light on some shared features of early ichthyopterygians, which will certainly serve as useful information to anyone working on this group in the future.

I am suggesting minor revisions and look forward to seeing the published form of the manuscript in the near future.

Best regards,

Robin Cuthbertson

·

Basic reporting

The manuscript is ver well written and clearly structures. The figures document the findings well, but could potentially be extended slightly (see below).

Experimental design

The investigation was conducted rigorously and throuroughly, the data is presented clearly and put into context of the current of knowledge on the topic.

Validity of the findings

Please see above

Additional comments

The mansucript “A new Lower Triassic ichthyopterygian assemblage from
Fossil Hill, Nevada” by Kelly et al. describes two new finds of Lower Triassic ichthyosaurian taxa from the Fossil Hill locality in Nevada, otherwise famously known for its Middle Triassic ichthyosaur fauna. The specimens, albeit incomplete, provide important new data points in the context of the early evolution and biogeography of lower Triassic ichthyosaurs in ecosystems that are still in the recovering phase after the Permian Triassic mass extinction.
The manuscript is well-written, clearly structured and presents important data. As such is very well suited for publication in PeerJ. I have only a few minor comments, which the authors may want to consider prior to publication. Please also see some minor comments added directly in the pdf file.

1. I think adding an additional figure with a global biogeographic map of the Lower Triassic indicating the localities at which the discussed ichthyoptergians have been found would be a useful addition and support the discussion on biogeography and potential dispersal routes.

2. Much of the description of the specimens is focused on the dentition – and rightly so as it bears the gros of characters. However, especially specimen USNM 559349 (cf. Utatsusaurus) consists of a nicely preserved mandible with portions of the dentary, splenial, and surangular. It may be worth adding a short paragraph on how the morphology and sutural pattern compares to Utatsusaurus or other lower Triassic taxa. Especially the large foramina visible in the photograph in Figure 2 are very prominent, but not mentioned in the text. Also, the authors mention an isolated tooth in this specimen, which is not shown in figure 2. It could be added as an extra panel to the figure.
Figure 2A should be rotated to show the mandible in anatomical orientation.

3. An additional close-up photograph of the dentition in Figure 3 would also be helpful.

---

## Round 0.2 · accepted · Accept

Thank you for your close attention to the comments on the previous version of the manuscript.